# Interventions to Improve Knowledge, Attitudes, and Uptake of Recommended Vaccines during Pregnancy and Postpartum: A Scoping Review

**DOI:** 10.3390/vaccines11121733

**Published:** 2023-11-21

**Authors:** Imen Ayouni, Edina Amponsah-Dacosta, Susanne Noll, Benjamin M. Kagina, Rudzani Muloiwa

**Affiliations:** 1Department of Paediatrics and Child Health, Red Cross War Memorial Children’s Hospital, University of Cape Town, Cape Town 7700, South Africa; rudzani.muloiwa@uct.ac.za; 2Vaccines for Africa Initiative, School of Public Health, Faculty of Health Sciences, University of Cape Town, Cape Town 7935, South Africa; edina.amponsah-dacosta@uct.ac.za (E.A.-D.); susanne.noll@uct.ac.za (S.N.); benjamin.kagina@uct.ac.za (B.M.K.)

**Keywords:** vaccine uptake, interventions, knowledge, attitudes, maternal vaccination, pregnancy, postpartum, influenza, pertussis, COVID-19, tetanus

## Abstract

Tetanus, pertussis, influenza, and COVID-19 vaccines are recommended for the prevention of related morbidity and mortality during pregnancy and postpartum. Despite the established benefits of vaccination for prenatal and postnatal women, maternal vaccination is not universally included in routine antenatal programs, especially in low- and middle-income countries. Furthermore, the uptake of recommended vaccines among pregnant and postpartum women remains below optimum globally. This review aimed to map the evidence on interventions to improve knowledge, attitudes, and uptake of recommended vaccines among pregnant and postpartum women. We conducted a comprehensive and systematic search for relevant literature in PubMed, Scopus, Web of Science, EBSCOhost, and Google Scholar. Overall, 29 studies published between 2010 and 2023 were included in this review. The majority (n = 27) of these studies were from high-income countries. A total of 14 studies focused on the influenza vaccine, 6 on the Tdap vaccine, 8 on both influenza and Tdap vaccines, and only one study on the COVID-19 vaccine. Patient-centered interventions predominated the evidence base (66%), followed by provider-focused (7%), health system-focused (10%), and multilevel interventions (17%). Overall, the effect of these interventions on knowledge, attitudes, and uptake of maternal vaccines was variable.

## 1. Introduction

Pregnant women, fetuses, and neonates are particularly vulnerable to infectious diseases. Infections, including those that can be prevented by vaccination, are associated with high morbidity and mortality among expectant mothers and their fetuses and neonates [1]. During the 1918 and 2009–2010 influenza A (H1N1) pandemics, for example, pregnant women and newborns were more susceptible to severe morbidity and mortality [1,2,3,4]. 

Newborns and young infants can be protected from some infections by the antibodies they receive from their mothers via transplacental transfer. Pregnant women who receive vaccinations, often known as vaccination in pregnancy (VIP), are the origin of these antibodies [5,6,7,8]. Vaccinating pregnant women has a dual advantage. The expecting mother will be protected from diseases to which she may be particularly susceptible when pregnant. Additionally, this protects the developing baby from congenital infections and other adverse effects of maternal transmission of diseases. Second, through the placental transfer of neutralizing immunoglobulin G (IgG) and secretory immunoglobulin A (IgA) in breast milk, vaccination during pregnancy may also protect infants from infections during the first few months of life [9]. In instances in which mothers did not receive recommended vaccinations before or during pregnancy, postpartum vaccination may also offer protection to infants during the few first months of life. Mothers in the postnatal period are less likely to expose their newborns to a virus or bacteria if they have developed immunity to them [10,11]. 

Tetanus, pertussis, influenza, and COVID-19 vaccines are recommended in every pregnancy and during postpartum if the mother was not vaccinated in the antepartum period [12,13,14,15,16]. The tetanus vaccine has provided the most extensive experience with VIP. In collaboration with the United Nations Children’s Fund and the United Nations Population Fund, the WHO developed the Maternal and Neonatal Tetanus Elimination Initiative in 1999 [17]. Maternal and neonatal tetanus has been eliminated in 47 of 59 “at risk” of maternal and neonatal morbidity and mortality due to increasing vaccination coverage among pregnant women and improving delivery cleanliness. 

Coverage of recommended vaccines remains suboptimal among pregnant women globally, especially in resource-constrained settings [18,19,20]. Lack of clear policies and guidelines, ineffective cold-chain management, and limited reporting and monitoring systems are barriers to vaccine delivery and uptake [21]. 

The benefits of VIP and during postpartum can only be realized if the recommended vaccines are universally accessible and optimally taken up. Maternal knowledge, attitudes, and beliefs about vaccines are important predictors of vaccine acceptance and uptake. Therefore, addressing immunization determinants, such as mothers’ knowledge, attitudes, and beliefs about maternal and childhood vaccines, is critical to increasing global vaccination rates and reducing global vaccine-preventable maternal and neonatal morbidity [3,22]. 

A recently published systematic review that aimed to explore the health systems’ determinants of maternal vaccine delivery and uptake in LMICs found that there is a lack of research that aims at promoting health and providing education during pregnancy, with a specific focus on preventing vaccine-preventable diseases (VPDs) [21]. Authors found that interventions on maternal immunization have been explored in high-income countries, but they have not received adequate attention in LMICs. Additionally, they highlighted the importance of gaining a deeper understanding of the role of policymakers, particularly National Immunization Technical Advisory Groups (NITAGs), and how they impact the delivery and uptake of maternal vaccines in LMICs. Therefore, maternal vaccination programs should prioritize building trust within communities and engaging key stakeholders as foundational components [21].

Optimal uptake of the recommended vaccines can be aided by several strategies such as educational interventions to patients and healthcare professionals, as well as health systems, and policy interventions.

It is critical to design innovative, comprehensive health systems-based interventions to enhance the delivery and acceptance of maternal vaccines in LMICs, especially when the crisis of vaccine hesitancy has deepened with the emergence of the COVID-19 pandemic. This has largely been driven by the “infodemic”, characterized by misinformation and disinformation, that accompanied the pandemic [23]. 

The objective of this scoping review is to map and describe interventions aimed at improving knowledge, attitudes, and uptake of recommended vaccines among pregnant and postpartum women globally. Where evidence of the impact of these interventions has been reported, these are further addressed in this review.

## 2. Materials and Methods

This scoping review was conducted in line with the PRISMA Extension for Scoping Reviews (PRISMA-ScR) checklist [24] provided in Appendix A. The primary objective of this review is to describe and map evidence on interventions to enhance knowledge, attitudes, and uptake of recommended vaccines among pregnant and postpartum women. The secondary objective is to describe the intervention’s effects. The interventions were classified per the authors’ reporting of the type of intervention.

### 2.1. Search Strategy

Two reviewers (IA and SN) performed a preliminary search for relevant literature through electronic databases and platforms, namely PubMed, Scopus, Web of Science, and EBSCOHost (Academic Premier, Africa Wide information, CINAHL, Health Source Nursing Academic, Medline, APA Psych, and APA PsycInfo). Supplementary searches for additional literature were conducted in Google Scholar and manual reference searches by checking the references listed in the included articles. 

The literature search was guided by a Boolean search strategy using the following key terms: “Pregnancy”, “pregnant women”, “expecting mother”, “postpartum women”, “antenatal”, “prenatal”, “postpartum period”, “vaccine uptake”, “vaccine acceptance”, “vaccine intention”, “vaccine hesitancy”, “intervention”, and “educational intervention”. Search strategies were tailored to the specific requirements of each database. Full search strategies for each database are provided in Appendix A. We did not apply any geographical or publication date limitations. The final search date was April 2023. 

### 2.2. Eligibility Criteria and Study Selection

The literature obtained through database searches was exported to the reference management software Endnote version 20 and then imported into the Rayyan systematic review management online platform for duplicate removal and screening of titles, abstracts, and full texts [25]. The title and abstract screening were guided by defined eligibility criteria. We only included the literature published in the English or French language. Studies adopting an experimental design, such as randomised controlled trials, quasi-experimental studies, before and after studies, and those presenting findings on the impact of an intervention(s) on the knowledge, attitudes, and uptake of recommended vaccines during pregnancy and postpartum, were eligible for inclusion. Observational studies and nonprimary quantitative and qualitative studies were excluded. Lastly, only peer-reviewed published papers were selected for inclusion in this review.

### 2.3. Data Extraction and Analysis

The first author (IA) screened the title and abstracts. Then, all eligible articles were assessed by reviewing the full text. Data were extracted from the included studies guided by a data extraction sheet designed for this review. The second author (EAD) verified screening for accuracy, and disagreements were resolved by consensus following discussions involving the authors BK and RM. The name of the author, publication year of the manuscript, study location, type of vaccine(s), study design and sample size, the intervention(s), and main findings were extracted (the data extraction sheet for this review is presented in Appendix A). The findings of this review are presented in tables and figures and are accompanied by a narrative, descriptive summary of the extracted data. 

## 3. Results

The initial search from the databases using the specified search terms yielded 907 potentially relevant articles. A total of 124 papers were excluded as duplicates, leaving 783 papers for title and abstract screening. Following that, 731 papers were excluded based on the eligibility criteria. A total of 79 articles were eligible for full-text screening. Upon full-text assessment, studies were further excluded because they did not adopt an interventional study design (n = 25), the study population did not include pregnant and/or postpartum women (n = 9), and study outcomes did not assess vaccination knowledge, attitudes, or uptake during pregnancy and/or postpartum (n = 16). Finally, 29 studies were judged to be eligible and included in this review (Figure 1). A summary of study characteristics and main findings is provided in Figure 2 and Figure 3, and Table 1.

### 3.1. Characteristics of the Included Studies

All papers were published between 2010 and 2023. Most of the included studies (52%) were conducted in the United States (n = 15) [27,29,30,32,33,34,35,42,44,48,49,50,53,54,55]. Three studies were conducted in Canada [31,36,51], two in Italy [38,41], and two in Taiwan [37,52]. The other seven studies occurred in Hong Kong [28], United Kingdom [39], France [47], India [46], Greece [40], Jordan [43], and Australia [45]. The majority of the studies (93%) were from high-income countries [27,28,29,30,31,32,33,34,35,36,37,38,39,40,41,42,44,45,47,48,49,50,51,52,53,54,55], and only two studies [43,46] were from low- and middle-income countries. In total, fourteen studies focused on the influenza vaccine [27,28,30,31,32,33,34,35,36,37,39,40,41,46], six studies on Tdap vaccine [42,45,47,48,49,52], eight studies on both Influenza and Tdap vaccines [29,38,44,50,51,53,54,55], and only one study was conducted on the COVID-19 vaccine [43]. The study population was pregnant women only in twenty studies [27,28,29,30,31,32,33,34,35,36,37,39,40,41,44,46,53,54,55], postpartum women only in seven studies [42,45,47,49,51,52,53], and included both pregnant and postpartum women in two studies [43,50]. In total, 55% of studies were randomized controlled trials [27,28,29,30,31,32,33,34,35,36,37,46,48,53,54,55], 31% of studies were pre–post interventions [38,39,40,41,42,47,49,50,52], and 14% of interventions had a quasi-experimental study design [43,44,45,51]. The majority of the interventions included in this review were patient-centered interventions (66%), and the other interventions were provider-focused (7%), health system-focused (10%), or multilevel interventions (17%). 

### 3.2. Main Findings from the Included Interventions Studies Aimed at Improving Knowledge, Acceptance, and Uptake of Maternal Vaccines

#### 3.2.1. Patient-Centered Interventions

Patient-centered interventions are individual-level interventions conducted on pregnant or postpartum women. Twenty of the included studies evaluated the impact of the intervention(s) on solely pregnant and/or postpartum women [27,28,29,30,31,32,33,34,35,36,37,38,39,40,41,42,43,44,45].

Most of those interventions (n = 13/19) were digital interventions [27,29,30,31,32,34,35,36,37,38,39,43,44]. The most used tool in the digital interventions was text messaging [27,31,32,35,45], and the majority of them did not find an effect of interventions using text messages, even those that tested positively (gain-frame) and negatively oriented (loss-frame) types of messages [27,35]. Only one intervention [32] that was conducted in the United States and used five weekly text messages showed an increase in the uptake of the influenza vaccine among pregnant women associated with the intervention. Another digital intervention that used long animation [32] showed an increased likelihood of vaccination against influenza among the study participants. Two other digital interventions used applications [34,37] and resulted in increased knowledge and improved attitudes, including intention to vaccinate against influenza among pregnant women. A study [34] that implemented a digital intervention consisting of exposing participants to either a website with information on vaccines and an interactive social media component or a website with vaccine information only found an intervention effect on the uptake of the influenza vaccine and no effect on the uptake of the Tdap vaccine among participants in the study. Another digital intervention [30] that used the video education method showed that the intervention positively impacted vaccination health beliefs without influencing vaccination uptake rates. One more educational digital intervention [44] that was conducted online in the United States showed that the uptake of influenza and Tdap vaccines increased among participants in the intervention group. Another intervention that focused on the COVID-19 vaccine [43] and consisted of a tele-education intervention that used multiple digital tools, such as interactive education phone sessions, phone calls consultancy, text messages, and digital education booklet, decreased their hesitancy and improved their willingness to be vaccinated against COVID-19.

Lastly, four other patient-focused interventions consisted of face-to-face education or counselling/training sessions [28,38,41,42] of participants and showed a moderate to high improvement in participants’ attitudes towards vaccination during pregnancy or postpartum and a moderate to significant increase in the uptake of influenza and/or Tdap vaccines. Another educational intervention [40] used a leaflet with information on the potential benefits of influenza vaccination during pregnancy; the leaflet was presented by their obstetrician, and participants had the opportunity to discuss vaccination during pregnancy with their obstetrician. This intervention showed a moderate increase in the uptake of the influenza vaccine among pregnant women. Finally, two more educational interventions [33,45] used pamphlets that increased the uptake rate of the influenza vaccine among pregnant women [33] and the Tdap vaccine among postnatal women [45].

#### 3.2.2. Provider-Focused Interventions

Provider-focused interventions were conducted either on healthcare providers or on health workers and conducted by them. Two of the included studies [46,47] used provider-focused interventions. The first one [47] consisted of doctors and midwives who provided oral and written information about Tdap to postpartum women, and it had a limited impact as it resulted in a small increase in the uptake rate among study participants. The second one [46] involved education and training to clinicians about influenza vaccination during pregnancy. Then, uptake rates per clinician were compared, and the intervention moderately increased the uptake rate.

#### 3.2.3. Health System Interventions

Health system interventions are implemented at any level of the health facilities and include different vaccine delivery models, a computer-based clinical decision support system, and an “opt-in” vs. standing order. Three health system-focused interventions are reported in this review.

First, an intervention [51] assessed and compared Tdap vaccine coverage among pregnant women between four province-based implementation models, which are as follows: existing standard practice, family medicine groups, obstetrics clinics, and oral glucose challenge test. The findings suggest that, compared with the standard practice, vaccine uptake rates were significantly higher when the Tdap vaccine was offered in family medicine groups and obstetric clinics providing antenatal care.

The second health system intervention [48] was a two-stage intervention that implemented an “opt-in” order as part of the preprinted postpartum orders. This required providers to check the order for both vaccinations to be given to women after delivery before hospital discharge. Following that, a standing order for Tdap vaccines for postpartum women instead of “opt-in” order was introduced, and a small increase in the uptake rate with the “opt-in” order and a significantly higher increase with the standing order policy were found.

The third intervention consisted of a computer-based clinical decision support system, which is an application that was integrated into the hospital information system [49]. This was a two-stage intervention: firstly, an “opt-in” order as part of the preprinted postpartum orders was implemented at the end of November 2009. This required providers to check the order for both vaccinations to be given to women after delivery before hospital discharge. Then the intervention simplified the delivery of vaccinations by implementing a policy with standing orders for postpartum vaccination for and seasonal and H1N1 influenza vaccination was implemented in February 2010. The standing orders empowered nurses to deliver influenza and/or Tdap vaccines without an additional order from the physician. Tdap would be administered unless the patient refused or had a contraindication to vaccination. The control hospital maintained standard practice. Randomly selected hospital charts of women after delivery were reviewed for receipt of Tdap and demographic data. Tdap vaccination rates among Tdap postpartum women were evaluated and a review of 1252 charts was conducted (648 intervention hospitals; 605 control hospitals) from women with completed deliveries. The intervention targeted postpartum women, and it was associated with a significantly increased uptake rate of the Tdap vaccine among study participants.

#### 3.2.4. Multicomponent and Multilevel Interventions

The remaining five studies included multicomponent or multilevel interventions [50,52,53,54,55] that targeted practices, providers, and patients.

A multicomponent multicentre intervention conducted in the United States [50] evaluated the effect of a multimodal intervention on the uptake rates of Tdap, HPV, and influenza vaccines in outpatient obstetrics and gynaecology clinics. In addition to this, after the intervention’s implementation, an order was processed for all indicated vaccines regardless of insurance. This study used evidence-based interventions largely implemented in other settings and showed increased rates of influenza, Tdap, and HPV vaccination in outpatient obstetrics and gynaecology clinics.

Another multilevel intervention [53] was conducted in the United States and aimed at increasing vaccination against influenza, Tdap, and HPV uptake among pregnant and nonpregnant women. Both intervention and control clinics showed improved vaccination among pregnant but not nonpregnant participants. However, there were not significant differences between the intervention and control groups. Another cluster randomized controlled trial [54], also conducted in the United States, consisted of a multicomponent intervention, and involved the identification of a vaccine champion, provider-to-patient talking points, educational brochures, posters, lapel buttons, and iPads loaded with a patient-centred tutorial. Despite the increased antenatal influenza and Tdap vaccination uptake in the intervention group, the increase was not statistically significant. This study demonstrated that the provider’s recommendation was the factor most strongly associated with actual receipt, regardless of study group or vaccine.

One more multicomponent intervention [52] was conducted in Taiwan and incorporated intensive physician and nursing educational training to provide the required knowledge of early-onset neonatal Streptococcus Group B (GBS) infection, neonatal pertussis infection, and perinatal preventive strategies for both GBS and pertussis to all obstetric physicians, clinic nurses and medical assistants. Then, an office-based intervention was conducted, which involved a pertussis education program in prenatal GBS screening clinics. Lastly, a prenatal Tdap education alert was incorporated into the electronic prenatal care medical record system, which consisted of an electronic reminder. In case a physician ordered vaginal and rectal swab cultures for possible GBS colonization for a woman without Tdap vaccination documentation, the antenatal Tdap education alert reminded healthcare providers with the following message: “Health education about Tdap vaccination recommended for postpartum women”. Tdap vaccination was more likely during the postintervention period compared with the preintervention period. This study showed that incorporating pertussis information into prenatal education for GBS prevention was beneficial and that prenatal GBS screening sessions represent an opportunity for healthcare providers to provide pertussis postpartum vaccination education to eligible pregnant women to improve the uptake of Tdap vaccination among postpartum women.

Finally, one more multicomponent interventional study [55] was included in this review and consisted of a cluster- and individually randomized controlled trial. The intervention consisted of a multilevel cluster- and individually randomized controlled trial. The intervention targeted practice-, provider-, and patient-level barriers to vaccine uptake. The practice-level intervention included the identification of a vaccination champion and the implementation of the Assessment, Feedback, Incentives, and Exchange program. Provider-level interventions included educational training based on five behavioral constructs and approaches. A copy of A Clinician’s Guide to Vaccine Safety was provided at each clinic. For the patient-level intervention component, a theory-driven individually tailored application called MomsTalkShots was developed. Participants in the patient-level intervention arm then received video messages tailored to address the identified knowledge gaps and concerns and baseline vaccine intention. Participants were provided with up to six videos depending on their specific concerns. This study found no statistically significant increase in Tdap or influenza vaccine uptake overall. Those who had no intention or were unsure about receiving the influenza vaccine while they were pregnant and received just the patient-level intervention were 61% more likely to receive the influenza vaccine than those in the control group. There was no statistically significant difference in vaccine uptake for either influenza or Tdap between the four study arms.

## 4. Discussion

In this scoping review, we explored the current state of research on the interventions that aimed to improve the knowledge, attitudes, and uptake of recommended vaccines during pregnancy and postpartum. We included all interventional studies without time limitations that were conducted on at least one of the recommended vaccines during pregnancy and postpartum, which are tetanus, diphtheria, Tdap, influenza, and COVID-19. Findings from our review showed that this body of research covers a diversity of populations, settings, interventions, and different vaccines. Our review included studies that have an interventional study design only. The geographical representation of study intervention settings within this literature favored high-income countries. Most of the studies (93%) were from high-income countries [27,28,29,30,31,32,33,34,35,36,37,38,39,40,41,42,44,45,47,48,49,50,51,52,53,54,55], and only two interventions were conducted in low- and middle-income countries [43,46]. Our results are in agreement with previously published systematic reviews and meta-analyses [57,58].

The majority of the interventions included in this review were patient-centered interventions (66%), and the other interventions were provider-focused (7%), health system-focused (10%), or multilevel interventions (17%). Most of the patient-centered interventions were digital interventions (63%), which had different effects depending on the digital tools used. The overall effect of the digital interventions was limited to moderate, and interventions that used text messages [27,31,32,35,45] had the smallest effect. Our findings were consistent with the results from a systematic review and meta-analysis [59] that showed digital interventions had a small and nonsignificant effect and found that text messages were less effective than other digital interventions. The remaining patient-focused interventions consisted of one-to-one education or training sessions [28,38,41,42] of participants, and they showed a moderate to the high improvement in participants’ attitudes towards vaccination during pregnancy or postpartum and a moderate increase in the uptake of either influenza or Tdap vaccines. Our findings confirm results from previously published systematic reviews and meta-analyses [58] that showed that several educational interventions, specifically, pamphlets, websites, and brief one-to-one educational interventions are effective in increasing influenza vaccination uptake among pregnant women.

Our review found two provider-focused interventions [46,47], which consisted of doctors and midwives who provided oral and written information about Tdap to postpartum women and education and training to clinicians about influenza vaccination during pregnancy. These interventions showed a small to moderate effect. In addition to that, our review included three health systems interventions [48,49,51] that found the vaccine uptake rate was significantly higher when the Tdap vaccine was offered in family medicine groups and obstetric clinics providing antenatal care with the standing order policy and when using a computer-based clinical decision support system.

Moreover, our results confirm the findings from another systematic review of interventions to improve the uptake of pertussis vaccination in pregnancy [60] which found that multicomponent or multilevel interventions were associated with limited to no effect on vaccine uptake rates. Only one multicomponent interventional study [50] that used evidence-based interventions largely implemented in other settings showed increased rates of influenza, Tdap, and HPV vaccination in outpatient obstetrics and gynaecology settings. Additionally, two multilevel and multicomponent studies included HPV vaccines added to Tdap and influenza vaccines [50,53]. Including education about HPV vaccination among pregnant women could be an important strategy to increase vaccine uptake and vaccine coverage among eligible populations [61].

The effect of the interventions included in this review differs by study setting, especially by country. As vaccination during pregnancy and postpartum is not included in routine antenatal and postnatal care globally, it is only in some high-income countries that routine influenza and Tdap vaccination is incorporated in antenatal and postnatal care. Also, barrier access impacts the study results as the availability of vaccines in the study sites increases the vaccine uptake rates. With regards to recommendations, interventions should be tailored to the context and social determinants of the local setting; therefore, we recommend that countries, especially LMICs, investigate the context-specific factors associated with the uptake of recommended vaccines during pregnancy and postpartum, and design and implement interventions that are suitable to the country’s context and the population’s characteristics [62,63,64]. In addition, differences in the findings between studies could be due to the differences in the methodology and the study design of the included studies.

Given the far-reaching potential of maternal immunization for both women and infants, several new vaccines designed specifically for use during pregnancy are currently in development. These maternal vaccines have the potential to modify the epidemiology of several infectious diseases in pregnant women and their infants, as well as to improve global maternal and neonatal health. The antenatal vaccines that are under development include vaccines against respiratory syncytial virus (RSV), group B streptococcus (GBS), and cytomegalovirus (CMV) [65,66,67,68]. Safe and effective maternal and routine childhood vaccines will only be effective if mothers choose to receive them and parents decide to vaccinate their children. Adding to policies and strategies tailored to local settings, addressing immunization determinants, such as mothers’ knowledge, attitudes, and beliefs about maternal and childhood vaccines, is critical to increasing national, regional, and global vaccination rates and reducing global vaccine-preventable maternal and neonatal morbidity [22,69,70,71].

### Limitations and Further Considerations

Our study has limitations despite the comprehensive search strategy. We may have missed important literature published in languages other than English and French. Additionally, as this was a scoping review, we did not conduct risk-of-bias assessments or use meta-analysis methodology to assess the quality of the studies and the effectiveness of the included interventions. Therefore, we could not fully assess the differences between the findings across studies. Furthermore, only peer-reviewed published papers were considered for this review.

## 5. Conclusions

This scoping review aimed to give a broad overview of the interventions implemented worldwide to enhance knowledge and attitudes towards recommended vaccines during pregnancy and postpartum, in addition to scaling up the vaccination uptake rates among prenatal and postnatal women. The review also briefly summarized the interventions’ category, population, and design, as well as their effects and limitations. Gaps in existing knowledge related to implemented strategies and interventions are mainly found in low- and middle-income settings and especially in African countries. We strongly recommend the development and design of contextually relevant educational interventions for pregnant and postpartum women and for all recommended vaccines, including COVID-19 and future vaccines that are under development on the African continent and in LMICs in general.

## Figures and Tables

**Figure 1 vaccines-11-01733-f001:**
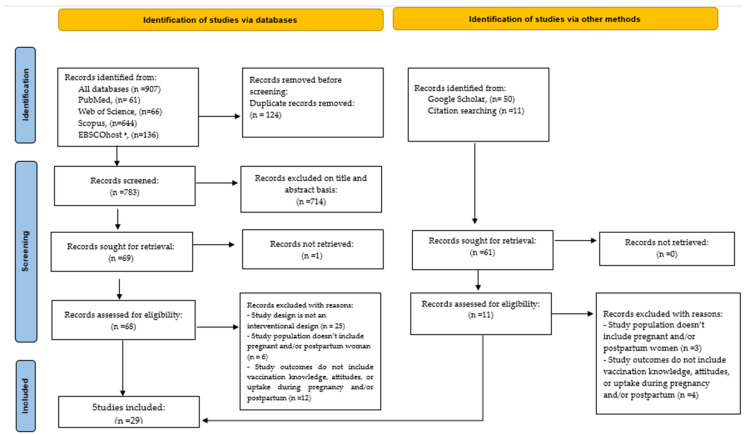
PRISMA flow diagram. Identification, screening, and inclusion of literature for this scoping review [26]. * Databases searched within EBSCOHost included Africa Wide information, CINAHL, Health Source Nursing Academic, Medline, APA Psych, and APA PsycInfo.

**Figure 2 vaccines-11-01733-f002:**
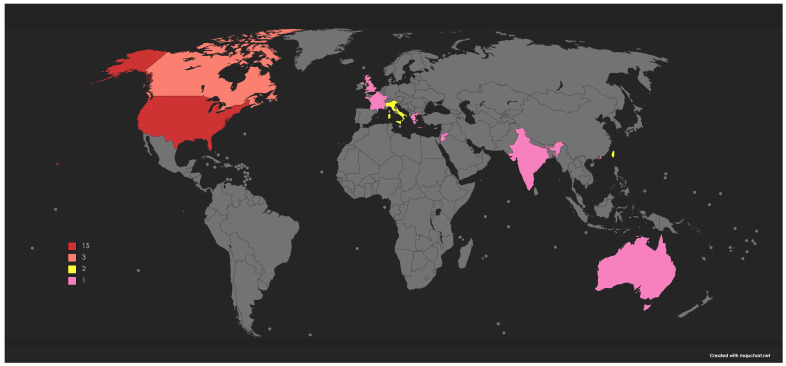
Number of intervention studies by country.

**Figure 3 vaccines-11-01733-f003:**
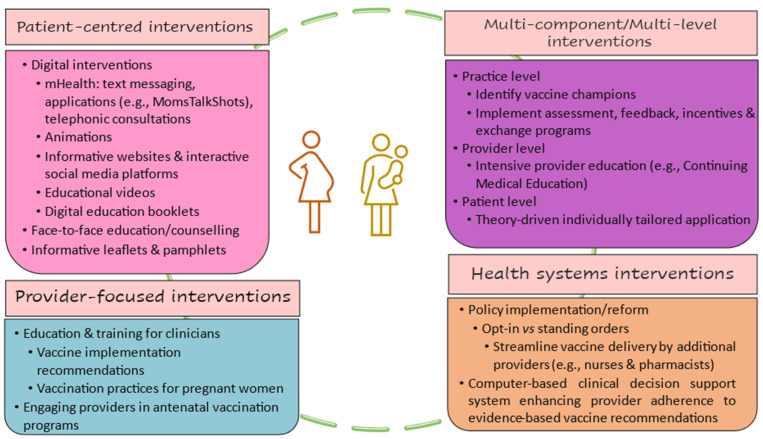
Different types of interventions included in this scoping review.

**Table 1 vaccines-11-01733-t001:** Summary of study characteristics and main results.

Author (Publication Year)	Country	Vaccine(s)	Population Group	Study Design	Category of the Intervention	Description of the Intervention	Main Results
Paula M. Frewa et al. (2016) [27]	United States	Influenza	Pregnant women	Randomized controlled trial (RCT)	Patient-centered intervention	Two forms of targeted persuasive messaging models: (i) affective messaging intervention (“Pregnant Pause” video) and (ii) cognitive messaging intervention (“Vaccines for a Healthy Pregnancy” video) in comparison to generic influenza vaccine information statements (VIS)	No effect after a single exposure to either affective messaging or cognitive messaging interventions on the vaccine uptake
Valerie Wing Yu Wong et al. (2016) [28]	Hong Kong	Influenza	Pregnant women	RCT	Patient-centered intervention	Brief, one-to-one education session on influenza vaccination uptake during pregnancy and the proportion of participants seeking out influenza vaccination	Uptake was higher among participants who received brief education compared to the standard care group. More participants in the education group-initiated discussion about influenza vaccination with their healthcare provider, but the difference was not statistically significant.
Sean T. O’Leary, (2019) [29]	United States	Tetanus–diphtheria–acellular pertussis (Tdap) and influenza	Pregnant women	RCT	Patient-centered intervention	Women were randomly assigned to one of three arms: “website with vaccine information and interactive social media components”, “website with vaccine information only”, or usual care.	Participants in both the first and second arms had higher vaccine uptake than the usual care group. There were no significant differences in vaccine uptake between study groups for the Tdap vaccine.
Kenneth Goodman et al. (2015) [30]	United States	Influenza	Pregnant women	RCT	Patient-centered intervention	Pre- and post-educational video on health beliefs was assessed, and unvaccinated women were subsequently interviewed by phone. Those in the control group viewed another video addressing handwashing hygiene.	The educational video positively influenced vaccination health beliefs without impacting vaccination uptake rates. The physician’s recommendation was strongly associated with the participant’s decision to vaccinate.
Michelle H. Moniz et al. (2013) [31]	Canada	Influenza	Pregnant women	RCT	Patient-centered intervention	Participants received 12 weekly text messages regarding general preventive health information in pregnancy and the importance of influenza vaccination during pregnancy.	Text messaging intervention was not effective at increasing influenza vaccination uptake rates among a low-income, urban, ambulatory pregnant population.
Melissa S. Stockwell et al. (2014) [32]	United States	Influenza	Pregnant women	RCT	Patient-centered intervention	Participants in the intervention group received five weekly text messages regarding influenza vaccination and two text message appointment reminders.	Text messaging was associated with increased influenza uptake, in a low-income obstetric population, mainly those who received the intervention early in their third trimester.
Pamela M. Meharry et al. (2013) [33]	United States	Influenza	Pregnant women	RCT	Patient-centered intervention	A pamphlet was tailored to pregnant women entitled “Influenza in Pregnancy”.	The pamphlet significantly increased the pregnant women’s perceptions of the safety and benefits of vaccination against influenza during pregnancy and the overall uptake.
Matthew Z. Dudley et al. (2022) [34]	United States	Influenza	Pregnant women	RCT	Patient-centered intervention	Educational videos through MomsTalkShots, algorithmically tailored application to pregnant women’s vaccine attitudes, including intentions	MomsTalkShots increased the perceived risk of maternal influenza infection and confidence in influenza vaccine efficacy.
Paula M. Frewa et al. (2014) [35]	United States	Influenza	Pregnant women	RCT	Patient-centered intervention	Two types of messages: positively oriented (“gain-frame”) messages communicate information by emphasizing the benefits of receiving the vaccine, and negatively oriented (“loss-frame”) messages emphasize the risks of not receiving the vaccine	Neither gain- nor loss-framed messages were significantly associated with an increased likelihood of influenza vaccination among pregnant women.
Mark H. Yudin et al. (2017) [36]	Canada	Influenza	Pregnant women	RCT	Patient-centered intervention	Two messages weekly for four consecutive weeks, reinforcing that vaccination against influenza is recommended for all pregnant women and is safe during pregnancy and breastfeeding	Weekly text messages did not increase the likelihood of getting vaccinated during pregnancy.
Ya-Wen Chang et al. (2022) [37]	Taiwan	Influenza	Pregnant women	Multicenter randomized controlled trial	Patient-centered intervention	An “Influenza Vaccination Reminder Application” was evaluated for improving vaccination intention among pregnant women.	The intervention statistically significantly increased pregnant women’s knowledge about influenza and vaccines, strengthened their positive attitudes towards maternal influenza vaccination, and promoted positive behavioral intention toward influenza vaccination.
Claudio Costantino et al. (2021) [38]	Italy	Influenza and Tdap	Pregnant women	Multicenter, pre- and post-educational intervention	Patient-centered intervention	Educational intervention on vaccination during pregnancy, immunization during life course, and vaccination recommended in Italy conducted by healthcare workers during childbirth classes	The educational intervention improved considerably the vaccination uptake during pregnancy.
Joanne Parsons et al. (2022) [39]	United Kingdom	Influenza	Pregnant women	Before and after interventional study	Patient-centered intervention	4 min long animation addressing beliefs about the risk of influenza and the efficacy of the vaccination.	An increased appraisal of the likelihood of getting flu during pregnancy and severity of influenza infection during pregnancy, and increased intentions to accept influenza vaccination during pregnancy
Helena C. Maltezou et al. (2019) [40]	Greece	Influenza	Pregnant women	Before and after interventional study	Patient-centered intervention	A leaflet that was given if the participant asked for it. It included information about the complications due to influenza infection among pregnant women and neonates and the efficacy and safety of influenza vaccine administered during pregnancy.	Educational intervention was associated with an increased uptake rate of 19.5% among pregnant women compared to <2% in the past years.
Stefania Bruno et al. (2021) [41]	Italy	Influenza	Pregnant women	Pre–post intervention study	Patient-centered intervention	Training sessions carried out during a birthing preparation course, aimed at increasing the attitude toward vaccination among pregnant women	Vaccination knowledge and attitude significantly increased after a training session.
Nutan B. Hebballi et al. (2022) [42]	United States	Tdap	Postpartum women	Before and after intervention study	Patient-centered intervention	A brief educational intervention session about maternal pertussis and the Tdap vaccine was given to interested hospitalized postpartum women, after which the Tdap vaccine was offered to eligible patients who did not receive it while they were pregnant. Medical records were reviewed to determine if surveyed participants received the vaccine before discharge.	A total of 25% were vaccinated before the study as part of routine hospital-based screening, and 38.2% were vaccinated after the intervention. Uptake increased with no significant difference before and after intervention.
Aaliyah Momani et al. (2023) [43]	Jordan	COVID-19	Pregnant and postpartum women	Quasi-experimental pre–post intervention study	Patient-centered intervention	Individual-centered tele-education (interactive education phone sessions, phone calls consultancy, text messages, and digital education booklet) was given to women in the intervention group for 2 weeks.	Education of pregnant women decreased hesitancy and improved willingness to be vaccinated against COVID-19.
Hallas Donna et al. (2023) [44]	United States	Influenza and Tdap	Pregnant women	Quasi-experimental intervention study	Patient-centered intervention	Study materials are provided online. The intervention was created to motivate participants to seek further information from scientific sources that were available to all study participants on the researchers’ website.	A total of 82% of vaccine-hesitant pregnant women had full prenatal vaccination coverage after receiving the intervention. The implemented intervention for vaccine-hesitant pregnant women was effective in shifting their status from hesitant to acceptor.
Elizabeth Helen Hayles et al. (2014) [45]	Australia	Tdap	Postpartum women	Quasi-experimental intervention study	Patient-centered intervention	Evaluate the role of message-framing vs. standard health information in the promotion of Tdap vaccination	Among susceptible mothers, 70% were vaccinated postintervention. No difference in vaccination rates, which were similar between groups. Overall pertussis vaccine coverage increased from 23% to 77%, and the ‘trusted’ environment with minimal access barriers had increased baseline pertussis vaccine coverage from 23% to 77%.
Joseph G. Giduthurim et al. (2019) [46]	India	Influenza	Pregnant women	RCT	Provider-focused intervention	Clinicians were interviewed and provided with antenatal influenza vaccination (AIV) implementation recommendations (global, academic, and local).	Engaging clinicians effectively reduced missed opportunities for AIV in urban middle-class settings. The absence of any similar impact in slum-based clinics might be the result of critical limitations of vaccine access.
C. Bonneau et al. (2010) [47]	France	Tdap	Postpartum women	Pre–post intervention study	Provider-focused intervention	Doctors and midwives received educational training on the benefits of pertussis vaccination during postpartum. Then, they provided oral (twice: midwife and doctor) and written information about pertussis and prescription for the Tdap vaccine for postpartum women.	Limited effect of the intervention At follow-up, vaccine uptake increased by 8%.
Sylvia Yeh et al. (2014) [48]	United States	Tdap	Postpartum women	Cluster randomized controlled trial	Health system intervention	A two-stage intervention: an “opt-in” order as part of the preprinted postpartum orders. Then, the intervention simplified the delivery of vaccinations by implementing a policy with standing orders for postpartum vaccination for Tdap and seasonal H1N1 influenza vaccination.	The introduction of the opt-in order achieved an increase in postpartum vaccination from 0% to 18%. The introduction of the standing order approach resulted in a further increase to 69%. No postpartum Tdap vaccinations were documented in the comparison hospital.
William E. Trick, et al. (2010) [49]	United States	Tdap	Postpartum women	Before and after intervention study	Health system intervention	Computer-based clinical decision support system incorporated into the hospital’s information system. When an order for iron supplementation was entered, a dialogue box was displayed containing a Tdap recommendation reminder, and an order for Tdap was generated and sent to the pharmacy and nursing staff unless the order was deselected.	The computer-based clinical decision support algorithm dramatically increased the Tdap vaccination uptake rate of postpartum women.
Sara E Mazzoni et al. (2016) [50]	United States	Influenza, Tdap and human papillomavirus (HPV)	Pregnant and postpartum women	Multicenter, pre–post multiple interventions	Multilevel intervention	1—Education sessions for non-provider medical staff on HPV and Tdap in pregnancy; 2—Existing standing orders for vaccines were revised or expanded depending on the vaccine; for instance, before the intervention, a standing order for a vaccine would be processed only if the vaccine was covered by insurance. After the intervention, an order for all indicated vaccines regardless of insurance; 3—Standing orders expanded to include influenza in the outpatient setting. For Tdap, each clinic began stocking and administering Tdap. Additional staff training, including providers, was conducted. Patient handouts were created and routinely given out at each first antenatal session and ultrasound visit.	The uptake rate of influenza vaccination increased from 35.4% in the preintervention period to 46.0% after the intervention. Tdap vaccination increased from 87.6% before the intervention period to 94.5% in the period after intervention.
Yinan Li et al. (2022) [51]	Canada	Tdap	Pregnant women	A quasi-experimental multicenter study	Health system intervention	Four province-based implementation models of maternal Tdap vaccine delivery: 1—existing standard of practice model, at local community service centers; 2—family medicine groups; 3—obstetrics clinic; 4— during the oral glucose challenge test (done during pregnancy to screen for gestational diabetes).	Compared with local community service centers, overall vaccine coverage was significantly higher when Tdap was offered in family medicine groups or an obstetrics clinic providing antenatal care. The oral glucose challenge test model did not improve overall vaccine coverage.
Po-Jen Cheng et al. (2015) [52]	Taiwan	Tdap	Postpartum women	Pre–post intervention study	Multilevel intervention	Intensive physician and nursing education programs about early-onset neonatal Streptococcus Group B (GBS) infection, neonatal pertussis infection, and perinatal preventive strategies for both GBS and pertussis, followed by an office-based intervention incorporating pertussis education programs into prenatal GBS screening clinics.	Tdap vaccination was more likely during the postintervention period compared with the preintervention period.
Sean T. O’Leary et al. (2019) [53]	United States	Influenza, Tdap and HPV	Pregnant women	Cluster randomized controlled trial	Multilevel intervention	Designation of vaccination champions, staff/provider training, assistance with vaccine purchasing/management, identification of eligible patients, standing order implementation, chart review/feedback, and patient education materials. Control practices continued usual care.	No significant differences between intervention and control groups for the uptake of influenza vaccine among pregnant women. Observed study arms increased their uptake. No significant differences in uptake of the Tdap vaccine in the intervention group vs. control groups.
A.T. Chamberlain et al. (2015) [54]	United States	Influenza and Tdap	Pregnant women	A cluster-randomized trial	Multilevel intervention	Identification of a vaccine champion, provider-to-patient talking points, educational brochures, posters, lapel buttons, and iPads loaded with a patient-centered tutorial.	Antenatal influenza and Tdap vaccination uptake was higher in the intervention group than in the control group, although not statistically significantly different.
Saad B. Omer et al. (2022) [55]	United States	Influenza and Tdap	Pregnant women	Cluster- and individually randomized controlled trial	Multilevel intervention	Obstetric clinics are randomized to receive the practice and provider-level interventions or continue their usual standard of care. The practice-level intervention: identification of a vaccination champion and implementation of the Assessment, Feedback, Incentives and Exchange program [56]. Provider-level interventions included Continuing Medical Education module. Patient-level intervention: theory-driven individually tailored application was developed.	No significant difference in vaccine uptake for either influenza or Tdap between the different study arms.

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
