# Peer review of "Interventions to Improve Knowledge, Attitudes, and Uptake of Recommended Vaccines during Pregnancy and Postpartum: A Scoping Review"

_vaccines, 2023, doi:10.3390/vaccines11121733_

Round 1

Reviewer 1 Report

Comments and Suggestions for Authors

This manuscript is potentially a useful update on and extension of previous systematic reviews of papers assessing the effectiveness of interventions to increase the uptake of influenza vaccines in pregnancy in that it includes a number of other vaccines, the post-partum period and more recent publications. However, I have several key methodological concerns. The manuscript is also very poorly written, with some sentences almost completely indecipherable, and with multiple typos. More detailed comments are below:

1. The authors state that they followed the PRISMA extension for scoping reviews. However, they did not follow the guidelines. Nor did they state where they did not follow the guidelines in the limitations section (another recommendation of the PRISMA guidelines). I have particular concerns about the following:

- The authors did not specify a clear research question. Instead they gave a vague research question that was not really worded as a research question. Thus it is not 100% clear what they were aiming to do.

- The authors did not describe having an 'a priori' protocol and did not show evidence of publishing this prior to commencing the review.

- The authors did not provide a date that the search was undertaken.

- Only a single author screened the search results, and only screened title and abstract. This is a critical issue. They did not describe how the screening was done beyond reading the title and abstract.

2. The table of findings seems to be ordered in a very ad hoc way, making it difficult to read and interpret. They should be ordered by intervention type and also by methodology, so that the reader can more easily read the findings. Also in the table, some provide a specific % change in the outcomes of the interventions, but others just say that the finding was an increase. The authors should provide as much information as possible.

3. The narrative expression of the findings was so poorly written it is almost incomprehensible, making it very difficult to read. It also makes very tedious reading. The authors need to summarise their findings in the narrative section so that the reader can better understand what they found.

4. There is an inaccuracy in the manuscript. In the discussion, the authors state that their findings confirm the findings from another systematic review on the interventions to improve uptake of pertussis vaccine in pregnancy. However, the review was about interventions to improve the uptake of influenza vaccine in pregnancy. It is difficult to see how the authors could have made such a mistake.

5. The discussion does not really discuss potential reasons for differences in findings between studies - that is why some studies found an intervention resulted in an increase in uptake and some others did not. The authors should discuss potential reasons for these differences.

6. There are errors in referencing.

Overall, the poor methodology and written expression of this study mean that the manuscript is not suitable for publication. The authors would need to redo their search and screening at the very least before the paper should be considered for publication as a scoping review.

Comments on the Quality of English Language

As above, the written English in this manuscript is sometimes almost completely incomprehensible. In other places there are numerous grammatical errors and typos. 

Author Response

Response to Reviewer 1

Comments and Suggestions for Authors

Point 1:

The authors state that they followed the PRISMA extension for scoping reviews. However, they did not follow the guidelines. Nor did they state where they did not follow the guidelines in the limitations section (another recommendation of the PRISMA guidelines). I have particular concerns about the following:

- The authors did not specify a clear research question. Instead they gave a vague research question that was not really worded as a research question. Thus it is not 100% clear what they were aiming to do.

- The authors did not describe having an 'a priori' protocol and did not show evidence of publishing this prior to commencing the review.

- The authors did not provide a date that the search was undertaken.

- Only a single author screened the search results, and only screened title and abstract. This is a critical issue. They did not describe how the screening was done beyond reading the title and abstract.

Response 1: We thank the reviewer for these comments. We did actually follow the PRISMA-ScR checklist for scoping reviews, however, we failed to include the PRISMA-ScR checklist in the supplementary materials. This has been adjusted and the PRISMA-ScR checklist is now included in the supplementary materials. Additionally, the research question/aim has been rephrased to make it clearer as follows: “The objective of this scoping review is to map and describe the interventions that are aimed at improving knowledge, attitudes, and uptake of recommended vaccines among pregnant and postpartum women that have been implemented worldwide. Where evidence of the impact of these interventions has been reported, these are further addressed in this review.”

Although publishing a protocol is always the best practice, a priori protocol has not been developed for the purpose of this review.

W actually mentioned the exact date when the search was undertaken. This was highlighted in the revised manuscript.

Finally, we failed to provide a detailed explanation of how the screening was conducted. In fact, four reviewers have been involved in the screening process, the first reviewer IA screened titles, abstracts, and full text for all eligible articles. The second author EAD assessed screening for accuracy. In case of disagreements, two other reviewers BK and RM were involved to resolve any disagreement. This has been reviewed and amended as follows:

“The first author (IA) screened the title and abstracts. Then all eligible articles were assessed by reviewing the full text. Data was extracted from the included studies guided by a data extraction sheet designed for this review. The second author EAD verified screening for accuracy and in case of disagreement authors to BK and RM for resolving inconsistencies and reaching an agreement.”

Point 2:

The table of findings seems to be ordered in a very ad hoc way, making it difficult to read and interpret. They should be ordered by intervention type and also by methodology so that the reader can more easily read the findings. Also in the table, some provide a specific % change in the outcomes of the interventions, but others just say that the finding was an increase. The authors should provide as much information as possible.

Response 2: The reviewer’s comment is well noted. We agree with the reviewer that the summary table should be reorganised, and findings should be re-ordered to make it easy to read. We made the changes to the summary table as suggested by the reviewer. However, we would like to draw the reviewer's attention to that as this is a “Summary table” which is included in the text, we only summarized the study characteristics and the main findings from the study, however, we did include in the supplementary materials another table (Supplementary Table 2) that describes the studies’ characteristics and contains detailed results from the articles.

Point 3: The narrative expression of the findings was so poorly written it is almost incomprehensible, making it very difficult to read. It also makes very tedious reading. The authors need to summarise their findings in the narrative section so that the reader can better understand what they found.

Response 3: We thank the reviewer for pointing this out. Language in the findings section was checked, and any mistake was corrected. The format was improved as well making the section into small paragraphs. As the aim of the review is to describe the interventions, a comprehensive description of the different studies should be provided for this research this section cannot be summarised further.

Point 4:  There is an inaccuracy in the manuscript. In the discussion, the authors state that their findings confirm the findings from another systematic review on the interventions to improve uptake of pertussis vaccine in pregnancy. However, the review was about interventions to improve the uptake of influenza vaccine in pregnancy. It is difficult to see how the authors could have made such a mistake.

Response 4: We thank the reviewer for this comment. We agree with the reviewer that this is a mistake in the referencing. This has been corrected and the reference has been replaced with the systematic review on the interventions to improve the uptake of pertussis vaccine in pregnancy.

Point 5: The discussion does not really discuss potential reasons for differences in findings between studies - that is why some studies found an intervention resulted in an increase in uptake and some others did not. The authors should discuss potential reasons for these differences.

Response 5: We thank you for your perspective. We agree with the reviewer that the potential reasons for differences between the findings of the studies are important and should be discussed. We would like to draw the reviewer’s attention that this point has been already discussed in the article in the discussion section, please find it in the following paragraph: “The effect of the interventions included in this review differs by study setting especially by country. As vaccination during pregnancy and postpartum is not included in routine antenatal and postnatal care globally, it is only in some high-income countries that routine Influenza and Tdap vaccination is incorporated in antenatal and postnatal care. Also, barrier access impacts the study results as the availability of vaccines in the study sites increases the vaccine uptake rates. With regards to recommendations, interventions should be tailored to the context and social determinants of the local setting, therefore we recommend that countries especially low- and middle-income countries investigate the context-specific factors associated with the uptake of recommended vaccines during pregnancy and postpartum, design and implement interventions that are suitable to the country context and population’s characteristics [61-63].” However, additional points have been added highlighting the difference between the methodology and the study design of the different interventions. In addition to that in the limitations section, we explained the limitations of not conducting quality appraisal and risk of bias assessment as the aim of this scoping review is to describe the different existing interventions.

Point 6: There are errors in referencing.

Point 6: Thank you for raising this point. References have been checked and the required changes have been applied.

Point 7: Overall, the poor methodology and written expression of this study mean that the manuscript is not suitable for publication. The authors would need to redo their search and screening at the very least before the paper should be considered for publication as a scoping review.

Response 7: We thank the reviewer for pointing this out. We agree with the reviewer that the search should be recent which is the case for our review as the search was conducted in April 2023.

Reviewer 2 Report

Comments and Suggestions for Authors

This is a fascinating and significant subject. I appreciate you bringing it up so it can be published.

The goal of this scoping review is to map the evidence that is currently available on interventions to enhance pregnant and postpartum women's knowledge, attitudes, and uptake of advised vaccinations globally. In addition to a scoping review, a systematic review of this subject is also possible. What evidence supports the writers' position?

The inclusion and exclusion standards were spelled out in detail.

Based on the classifications produced, the results and discussions were provided and the figure made the results easier to understand. 

Author Response

Response to Reviewer 2

Comments and Suggestions for Authors

This is a fascinating and significant subject. I appreciate you bringing it up so it can be published.

The goal of this scoping review is to map the evidence that is currently available on interventions to enhance pregnant and postpartum women's knowledge, attitudes, and uptake of advised vaccinations globally. In addition to a scoping review, a systematic review of this subject is also possible. What evidence supports the writers' position?

The inclusion and exclusion standards were spelled out in detail.

Based on the classifications produced, the results and discussions were provided and the figure made the results easier to understand.

Response: We thank the reviewer for these attentive and positive comments. We agree with the reviewer that a systematic review of this subject is possible. We actually planning on conducting a systematic review after publishing the scoping review.

Reviewer 3 Report

Comments and Suggestions for Authors

Thank you for the opportunity to review this manuscript, which I found very interesting for the topic chosen. I found the work well structured in terms of methodology and exposition. The methods and results are clearly stated. What I find the major flow of this paper is the fact that it claims to explore changes in knowledge, attitudes and vaccine uptake brought by the improvement interventions, but in fact the results only speak of changes in vaccine uptake and coverage. This may be a sufficient outcome for the article, but in this case the aims need to be reshaped. Other notes: figure 1 is of low quality; what is the rationale for the order in which the studies included in the first table are listed? There are several minor spelling and editing errors to be corrected.

Author Response

Response to Reviewer 3

Comments and Suggestions for Authors

Thank you for the opportunity to review this manuscript, which I found very interesting for the topic chosen. I found the work well structured in terms of methodology and exposition. The methods and results are clearly stated. What I find the major flow of this paper is the fact that it claims to explore changes in knowledge, attitudes and vaccine uptake brought by the improvement interventions, but in fact the results only speak of changes in vaccine uptake and coverage. This may be a sufficient outcome for the article, but in this case the aims need to be reshaped. Other notes: figure 1 is of low quality; what is the rationale for the order in which the studies included in the first table are listed? There are several minor spelling and editing errors to be corrected.

Response: We thank the reviewer for these attentive and positive comments. We have made improvements to Figure 1 in terms of quality. We agree with the reviewer that the summary table should be reordered. Amendments to Table 1 have been made as suggested by the reviewer.

We would like to draw the reviewer’s attention that our results do actually show changes in knowledge and attitudes. Would you please find below some of the results:

“. Another digital intervention that used long animation [31] and showed an increased likelihood of vaccinating against Influenza among the study participants. Two other digital interventions used applications [33, 36] and resulted in increased knowledge and improved attitudes including the intention to be vaccinated against Influenza among pregnant women.”

“Another digital intervention [29] that used video education method showed that the intervention positively impacted vaccination health beliefs without influencing vaccination uptake rates.  “

“One more educational digital intervention [43] that was conducted online in the United States showed that the uptake of Influenza and Tdap vaccines increased among participants in the intervention group. Another intervention that focused on COVID-19 vaccine [42] consisting of a tele-education that used multiple digital tools such as interactive education phone sessions, phone calls consultancy, text messages, and digital education booklet decreased their hesitancy and improved their willingness to be vaccinated against COVID-19.

Lastly, four other patient-focused interventions consisted of face-to-face education or counselling/training sessions [27, 37, 40, 41] of participants and showed a moderate to high improvement in participants' attitudes towards vaccination during pregnancy or postpartum and a moderate to significant increase in the uptake of Influenza and/or Tdap vaccines.”
